# Evaluation of the Recipe Function in Popular Dietary Smartphone Applications, with Emphasize on Features Relevant for Nutrition Assessment in Large-Scale Studies

**DOI:** 10.3390/nu11010200

**Published:** 2019-01-19

**Authors:** Liangzi Zhang, Eline Nawijn, Hendriek Boshuizen, Marga Ocké

**Affiliations:** 1National Institute for Public Health and the Environment (RIVM), 3721 MA Bilthoven, The Netherlands; liangzi.zhang@wur.nl (L.Z.); hendriek.boshuizen@wur.nl (H.B.); 2Division of Human Nutrition, Wageningen University & Research, 6708 PB Wageningen, The Netherlands; eline_nawijn@hotmail.com

**Keywords:** diet apps, recipe calculations, nutrient retention, food record, dietary intake assessment, technological innovations

## Abstract

Nutrient estimations from mixed dishes require detailed information collection and should account for nutrient loss during cooking. This study aims to make an inventory of recipe creating features in popular food diary apps from a research perspective and to evaluate their nutrient calculation. A total of 12 out of 57 screened popular dietary assessment apps included a recipe function and were scored based on a pre-defined criteria list. Energy and nutrient content of three recipes calculated by the apps were compared with a reference procedure, which takes nutrient retention due to cooking into account. The quality of the recipe function varies across selected apps with a mean score of 3.0 (out of 5). More relevant differences (larger than 5% of the Daily Reference Intake) between apps and the reference were observed in micronutrients (49%) than in energy and macronutrients (20%). The primary source of these differences lies in the variation in food composition databases underlying each app. Applying retention factors decreased the micronutrient contents from 0% for calcium in all recipes to more than 45% for vitamins B6, B12, and folate in one recipe. Overall, recipe features and their ability to capture true nutrient intake are limited in current apps.

## 1. Introduction

When assessing the dietary intake of a large population, an accurate dietary assessment plays a fundamental role [1]. Self-report dietary assessment methods, such as 24-hour dietary recall (24HDR), dietary record (DR), and food frequency questionnaire (FFQ), are commonly used to assess food consumption at both individual and population level [2]. Since underreporting, overreporting, misreporting, and interviewer bias can occur in those methods [3,4,5], assessing dietary intake with a high level of accuracy continues to be a major challenge in nutritional epidemiology and monitoring [6,7]. Moreover, cumbersome procedures of collecting details of foods are time-consuming and are associated with a high burden for both the respondent and the researcher [8]. This is especially the case for 24HDR and DR, which are open methods, and for which repeated measurements are needed to estimate usual dietary intake [9]. The burden laid on respondents can also lead to a low response rate, which may lead to bias in the survey results and diminish the representativeness of the sample [10].

Progress in Information and Communication Technology (ICT) in the past few decades has led to investigations into innovative strategies to overcome drawbacks of traditional pen-and-paper and interviewer-based dietary assessment methods [11,12]. One such innovative strategy is the use of mobile applications (apps) on smartphones for a dietary record. In the last decade, an increase in the number of smartphone users has led to a proliferation of mobile applications (apps) [13]. A popular category within all these apps are the health and fitness-related apps [14], mostly aimed at supporting dietary change and weight management [15,16]. Those apps usually include a food diary function, in which users can record the foods consumed and the consumed quantities. Apart from searching in a pre-defined food and beverage list and selecting pre-defined portion sizes [17], various features are available to help identify consumed foods, estimate portion size, and decrease the burden of food entering. Examples of those features are image-based food recognition and barcode scanner. Their potential on reducing the respondents’ burden, decreasing the effort of multiple self-administrations and on improving food recording accuracy have been investigated in both experimental and observational epidemiological studies, and have shown some promising results [6,18]. However, the knowledge on the performance of other specific features is still limited [19]. 

One feature of food diary apps is the recipe function for entering mixed dishes prepared at home. These are dishes consisting of multiple foods, with specific food preparation and often with cooking involved. For user-friendliness, the recipe function should be structured in a way that could easily guide the users in recording necessary information of a recipe. It should be able to assess the recipe intake of an individual, while mixed dishes are often prepared for more than one person [19]. Furthermore, for a better estimation of nutrient intake, an accurate recipe calculation should take nutrient loss of ingredients during cooking and food processing into account [20]. 

Some food diary apps have introduced a recipe function through the recent years [21,22]. The effectiveness of these recipe functions in capturing the food consumption and nutrient intake has not been fully evaluated. Moreover, the question whether the features of available recipe functions are also appropriate for dietary assessment as part of large-scale studies remains unanswered. Therefore, the aim of this study was to make an inventory of recipe function features in apps that could facilitate the estimation of nutrient intake of a large population. Furthermore, another aim was to evaluate the accuracy of the recipe function in capturing nutrient intake of popular dietary assessment apps by comparing their nutrient calculation with a standard calculation procedure.

## 2. Materials and Methods 

The starting point for app selection was an identification of dietary assessment smartphone apps in the Health & Fitness category of iTunes App Store and Google Play Store in the Netherlands between 15 and 23 October 2016. This selection was performed by Maringer et al. [20] and resulted in the identification of 176 dietary assessment apps. Further screening was performed in August 2017. Inclusion of a subselection of apps for this study required the app to meet the following criteria: (1) user rating >3 in iTunes App Store and Google Play Store, (2) user rating count >500 in iTunes App Store and Google Play Store, (3) >10,000 downloads in the both stores, (4) a recipe function which was freely available, actually present and functional. A recipe function was defined as “a functionality in which the user can create a mixed dish by entering and specifying the amount of each ingredient within the dish” [23,24]. Each app underwent initial screening based on descriptions and associated images in the app stores to check for the presence of a recipe function. Apps were downloaded onto a OnePlus 3T smartphone running Android 7.1.1 and a Huawei Mate 8 running EMUI 5.0.1 for analysis. The apps were checked manually to confirm whether a recipe function was freely available, actually present, and functional. Basic descriptive information about the apps was identified, such as app name, version number, operating platforms, number of installs, ratings, whether they can synchronize with their website, and country of origin. Subsequently, the recipe function of the selected apps was evaluated. 

To our knowledge, no widely accepted standard evaluation of the quality of the recipe function of apps exists. Therefore, a criteria list was made for evaluating features in the individual recipe function of apps. For each feature on the criteria list a rubric of assessment was created with a 1 (low)–5 (high) scoring scale. The criteria list and assessment rubric were modified upon findings from a pilot scoring and feedback from two nutritionists and three dietitians with different specializations. The criteria list and assessment include the following aspects of creating an individual recipe: options in searching ingredients, ways to record relevant information of the recipe, whether raw or cooked ingredients could be selected, consumed amount for both ingredients and the whole recipe, energy and nutrient expression, and whether the recipe could be saved and edited later (Table 1). Two researchers scored all the selected apps according to the criteria list independently. Inconsistent scores among the two researchers were discussed to reach agreed final scores. For scoring the criterion whether both raw and cooked foods are available in the food list, nine foods from the three most frequently used Dutch recipes (explained in next paragraph) were entered in each app (kale, potato, milk, mushroom, onion, salami, beef, pepper, and tomato). 

To be able to evaluate the accuracy of energy and nutrient content estimations, three recipes were entered into the individual recipe function of each app. The selection of recipes was performed by exploring the most frequent reported recipes in the Dutch diet using the data of the Dutch National Food Consumption Survey (DNFCS) 2007–2010 [25]. Three recipes with different preparation methods, like stewing, baking, and frying, were chosen from the twenty most frequently consumed recipes. The chosen recipes were boerenkool stamppot (mashed potato with kale), pizza with salami, tomato, and mushrooms, and hachee (a traditional Dutch stew based on beef and onions). Raw ingredients of the recipes were entered in the selected apps and a set of rules for entering ingredients were followed, in case the exact match of food items or amount indications could not be found across apps. If available, energy, macro- and micronutrient values of the recipe were obtained based on the displayed nutrient content in the app. For those apps where the nutrient contents were not shown at the recipe level, values from ingredients of a recipe were added up by researchers. Then, nutrient contents from the apps were compared with nutrient contents derived from the Dutch food composition database (NEVO) [26]. To account for nutrient loss due to cooking, retention factors suggested by the European Food Information Resource [27] were applied to the nutrients derived by NEVO, see complete calculation in Appendix A. A retention factor larger than 0 and lower than 1 implied nutrient loss due to cooking. A retention factor of 1 was used for energy and macronutrients for all ingredients in all recipes since they were not easily affected by cooking. Next to energy and macronutrient, micronutrients such as sodium, potassium, vitamin A represented as retinol equivalent (RE), vitamin C, calcium, vitamin E, vitamin B1, vitamin B2, vitamin B6, vitamin B12, and folate were selected for comparison between apps and the reference measure. Of these, sodium, potassium, and vitamin E had a retention factor of 1 for all ingredients in the three recipes mentioned above, hence, were deleted from analysis. Calcium also had a retention factor of 1, but was maintained in the analysis as an example. 

General characteristics of the 12 evaluated dietary assessment apps with recipe function were summarized. For each app, the mean score and standard deviation over all nine criteria was calculated (see Table 1). The mean and standard deviation of scores across apps were calculated for each criterion. Energy and nutrient content estimations of the three recipes for each app were analyzed using descriptive statistics. For nutrients with retention factor of 1, a direct comparison could be made with the nutrient contents derived from NEVO combining nutrient contents of raw ingredients in the appropriate amounts. For the micronutrients with retention factors below 1, the reference was the NEVO nutrient contents of the raw ingredients after applying the relevant retention factors. For showing the effect of the retention factors, a comparison with NEVO nutrient contents of raw ingredients without applying retention factors was also made. A difference in values between apps and the reference of more than 5% from the Daily Reference Intake (DRI) for adults was considered out of range [28].

To visualize the correlation between apps and nutrients, a principal component analysis (PCA) was conducted for each recipe separately with energy and macronutrients divided by their DRIs being set as variables. The first two principal components represent the most variation. This was done for energy and macronutrients only, since only 3 apps showed information on absolute amounts of micronutrients. The descriptive statistics were calculated using Excel 2016 software and the PCA was conducted in R version 3.5.0 (The R Foundation for Statistical Computing, Vienna, Austria). 

## 3. Results

### 3.1. App Selection

The starting point was a selection of 176 popular dietary assessment apps with food recording and available in English identified by Maringer et al. [21]. Then, apps were further narrowed down, with inclusion criteria of a user rating >3 in the iTunes App Store and Google Play Store, a user rating count > 500 in iTunes App Store and Google Play Store, >10,000 downloads in the Google Play Store, and a claimed recipe function in the app description. After manually checking for the presence of an individual recipe function in 30 included apps, 17 apps were excluded from further evaluation because of dysfunction of the app, the absence or dysfunctionality of a recipe function, or the inability to use the app due to requirements of a membership. After final exclusion of one app with a non-functioning individual recipe function, a total of 12 apps (21% of 57) were selected for evaluation in detail (Figure 1).

General characteristics of the remaining 12 apps can be found in Table 2. All apps operated on an Android platform, whereas IOS ranked as the second most-prevalent platform (10 apps). The highest number of installs was 50 million with 1844 thousand ratings for MyFitnessPal, the lowest was 100 thousand installs and 2000 ratings for Nutracheck. The rating scores among the apps ranged from 4.2 to 4.6 with the maximum score of 5.0. Four apps were made by US companies, two apps were made in Germany, and the rest of apps were made in other countries, mostly northwest Europe.

### 3.2. Qualitative Recipe Function Assessment

Agreed scores given to recipe functions of each app are shown in Table 3. Mean overall score of both apps and criteria was 3.0 (out of 5.0). The app Calories! had the highest score for its recipe function with an average score of 3.9 however, in contrast, Calories! had a rating score and number of installations at the lower range compared to other apps (Table 2). MyPlate and Health Infinity, on average, had the lowest scores of 2.2 and 2.3, respectively. 

The apps that had relative higher popularity, such as MyFitnessPal, Lose It!, Lifesum, and MyPlate, did not have any criterion that scored 5, while Calories! was achieved a score of 5 three times. Health Infinity scored 1 most often (three times) compared to other apps. 

Specifically, most of the evaluated apps could save a self-created recipe and edit it later, hence, this criterion ranked the highest (mean = 4.3) compared to other criteria. None of the apps included reminders for frequently forgotten ingredients, therefore, all apps scored 1 for that criterion. The available options that existed for searching ingredients for recipes included text search, barcode scanning, voice record, recent/frequent/saved food, create new food, choose from categories, and choose from a list of all food in alphabetic order. The number of options ranged from 2 to 6, where half of the apps had only 2 to 3 options, while only Nutracheck had all 6 options. The most frequently adopted options were search in a textbox and barcode scanning. FatSecret and Virtuagym Food had four searching options for food entering, but only two options for adding ingredients to recipes. In terms of options in searching raw or cooked foods, nearly all apps had both raw and cooked options for all or at least some foods in their dataset (mean = 3.3). An exception was The Secret of Weight, where, for the most foods, the text indicated raw while the picture showed cooked foods. In terms of indicating consumed amount in both ingredients and recipes, in Calories!, one could manually add a new serving unit to ingredients but not in recipes whereas, in Virtuagym Food, this was the other way around. Health Infinity had no options to chooe the amount of recipe consumed (scored as 1), and had only one built-in option when choosing the amount of ingredients. In terms of macronutrient information, Calories! was the only app that had energy and macronutrients expressed as both absolute amounts (mg, µg, etc.) and % of Recommended Daily Allowance (RDA). Most apps had energy and macronutrients shown only in absolute amounts. Since only four apps showed micronutrient for recipes, the average score for micronutrient availability ranked the second lowest with a score of 2.7. Among the apps with micronutrients, Calories! and MyNetDiary had both absolute amounts and % RDA for more than six micronutrients, while Virtuagym Food had only actual amounts. MyFitnessPal had only % RDA of less than six micronutrients.

### 3.3. Accuracy of Energy and Macronutrient Content Estimations

The differences in energy and macronutrient content estimations of the three recipes between the 12 popular dietary assessment apps and the value derived from NEVO are presented in Table 4. Macronutrient contents for both recipes and ingredients were not available in The Secret of Weight. Heterogeneity in differences was observed between recipes and between nutrients. Pizza had fewer differences >5% (*n* = 7) in the DRI as compared to boerenkool stamppot (*n* = 10) and hachee (*n* = 12). Carbohydrates (*n* = 2) and energy (*n* = 3) contents had fewer differences >5% in the DRI than protein (*n* = 13) and fat (*n* = 11). In total, around 20% of the differences were >5% DRI. Most apps underestimated the macronutrient content in boerenkool stamppot and pizza, while this was not observed in hachee. 

With 7 out of 12, Nutracheck had the most discrepancies >5% in the DRI compared to the reference, mainly caused by a discrepancy in fat and protein contents. YAZIO and Lifesum only had one difference of more than 5%. Health Infinity had lower protein contents in all three recipes, whereas Lose It! had lower fat in all three recipes. Virtuagym Food and YAZIO had similar patterns in all recipes, and both had lower fat in hachee as outliers. MyNetDiary had all macronutrients being out of range once, including a lower carbohydrate, lower protein, and higher fat in three recipes, respectively. In Figure 2, apps are plotted against the first and second principal component of all differences in macronutrient contents. Macronutrients plotted further from the center indicate a larger variance. Apps situated in the same direction with a certain nutrient indicate an overestimation of the nutrient and vice versa. Nutracheck laid outside compared to other apps for all three recipes. MyFitnessPal was the only app without discrepancies of more than 5%. Therefore, it was located around the center of the graph in all three recipes. 

### 3.4. Accuracy of Micronutrient Content Estimations

The micronutrient contents were analyzed for MyNetDiary, Calories! and Virtuagym in which it was available. The differences in micronutrient content estimations of the three recipes between the three popular dietary assessment apps, the micronutrient calculated from NEVO values in raw foods and the reference where retention factors was applied to NEVO are presented in Table 5. For most micronutrients except calcium, applying retention factors resulted in lower micronutrient levels than micronutrient levels in raw ingredients. The relative differences between the reference and using NEVO without applying retention factors ranged from 0% for calcium in all recipes, vitB12 in stamppot and vitB2 in hachee to more than 45% for vitamins B6, B12 and folate in hachee. Over the 3 recipes, 8 out of 24 differences (33%) were relevant (>5% of DRI) in case of a high content and high vulnerability of these nutrients of raw ingredients in a certain recipe. The relatively large difference in vitamin B6 and B12 in Hachee can be explained by the sensitivity to heat and the two cooking procedures in this recipe, i.e. frying and stewing. Whereas, boerenkool stamppot (*n* = 5) had more relevant differences than the other two recipes (*n* = 1 and 2 respectively), due to its high contents of vitamin C, vitamin A, vitamin B1, vitamin B6 and folate even if the retention factor was not so different from 1 (for example, vitamin A with a retention factor of 0.9). 

A larger proportion of difference >5% DRI was found in micronutrients (49%) than in energy and macronutrients (20%) when compared with the reference values. Among the three apps, MyNetDiary showed more differences > 5% DRI (*n* = 14 out of 24) than the other two apps (Virtuagym *n* =10, Calories! *n* =11) when comparing micronutrient values with the reference. In contrast to macronutrient comparisons, apps more often overestimated the contents of micronutrient in the recipes. The number and extent of overestimations were slightly larger when comparing with the reference than comparing with NEVO without applying retention factors, since the retention factors resulted in lower micronutrient contents in the reference values. The proportions of relevant differences found after comparing the apps to NEVO with or without applying retention factors were rather similar (49% vs. 51%), illustrating that in many cases the effects of differences in nutrient databases were much larger than differences due to applying retention factors.

## 4. Discussion

The current study evaluated the recipe function that was available in only one-fifth of the popular available food diary apps. We found a varying quality of recipe features across selected apps which were, on average, judged as suboptimal from research perspectives. Furthermore, capturing the true nutrient intake of mixed dishes is a challenge for this innovative dietary assessment method. A comparison of energy, macro-, and micronutrient contents of recipes between apps with a reference standard recipe calculation showed variation in terms of their ability to accurately estimate nutrient contents. In only three apps was micronutrient information available for recipes, and none of these apps included a procedure to take nutrient losses due to recipe processing into account, and the variability in micronutrient content databases was large. 

This is the first study to evaluate the recipe function of current popular dietary assessment apps in a standardized way in which the quality assessment was performed using a rubric of assessment which was made prior to the evaluation. The scores of recipe function were discussed by two researchers, which has eliminated mistakes and the bias of scoring. From the quality assessment of the recipe functions, apps were given a mean overall score of 3.0 (out of 5.0) where the highest score was 3.9 and the lowest 2.2. No correlations were found between the scores given in this study and the popularity and user ratings in app stores. This could illustrate that the recipe function was not the main aspect contributing to users’ overall app-experiences, or that researchers and users have different needs for dietary apps [9]. Some simplified features might be favored by users since it was observed that the user’s time invested for understanding and learning about an app should be small to sustain long-term app usage [30], whereas researchers are more concerned with features that could enable detailed and accurate data collection. This preference gap between the app users and researchers is important to select suitable features to be included in dietary assessment tools for large nutrition monitoring studies. 

Although the quality of recipe function in popular apps was not investigated before [13], several features of a recipe function were investigated by others since they are also relevant for recording food intake. In terms of options for searching ingredients in apps from the current study, all apps had a text searching option and the majority of the apps had a barcode function. Barcode scanning has been shown to save time and was favored by users in recording branded food items, however, the resulting nutrient intake estimation depends largely on the quality of the underlying food composition database within the app [31]. An aspect in which these apps differ from many web-based tools is that most of them do not have portion images, which may due to limited space in the user interface. Previous research has found that the incorporation of portion images was preferred by all age groups [9]. However the overall advantage of using portion images remains unknown [17]. In terms of nutrient information, the energy and macronutrient information was more complete in apps than micronutrient information, and this complied with the fact that energy and macronutrients were more closely correlated with weight change, which was the aim for most apps.

Features specific for creating recipes were evaluated. For instance, in addition to other basic features for entering recipes (i.e., add a name, ingredients, and serving number of the recipe), half of the evaluated apps had the capability to enter a photo and cooking explanation. However, this information was not used by the app to estimate nutrient intake. A photo of the recipe could help identify and estimate the amount of food consumed by participants, and could also reduce the extent of underreporting, especially for people with low literacy levels [17], while a cooking explanation provided information relevant for nutrient retention estimation. However, with the extra efforts required in using these features, they might be practical only in small-scale studies. Unlike computer/web-based dietary assessment tools for research purposes [32], all apps lack reminders for frequently forgotten ingredients when creating recipes (e.g., oil, spices, sugar, etc.), which may have partly contributed to the systematic underestimation of macronutrients in most apps found in other studies [33]. Also, current apps did not have pre-defined recipes that could be adapted by users whereas, in some computer-based software, standard recipes could be adapted by switching ingredients or changing the amount of ingredients [32]. However, the practicality of above features to be included in apps or to be used by participants, without the help of researchers, remains questionable. As a simpler alternative, the feature for saving frequently consumed or favorite foods in current apps was shown to save the efforts of users from entering the same recipes repeatedly and searching for food in a comprehensive food list [34]. 

In the present study, differences in energy, macro-, and micronutrient contents were found between the apps and the reference measure, which could be explained by several reasons. There were substantial differences in the nutrient contents of the recipe ingredients between apps, showing the differences in underlying nutrient databases. Apps were made by companies from different countries and they might have incorporated a nutrient database from their own countries which might have varying nutrient contents for certain foods, due to different cultivating environments [35]. Another source of nutrient values might be input from the app users. This has the benefit of customization of food consumed, however, has shortcomings in the accuracy of nutrients and can lead to quality losses in the food database [14]. 

The inability to enter exactly the same ingredients across the apps and the limited choice of food amounts may additionally explain part of the variation in nutrient estimation [33]. For example, it was difficult to find an exact match of beef steak in hachee, since there was a large variety of beef steak in different apps, and food amounts in grams were not available in some apps. However, for most other recipe ingredients, this problem did not occur. For micronutrients, the difference was also due to applying retention factors to the reference nutrient values, whereas all apps came up with the nutrient content of recipes by simply adding up the nutrient content of each ingredient without taking nutrient retention into account. 

Variations of nutrient content of three recipes between apps and the reference measure were observed in the present study, with fewer variations in energy and macronutrient than in micronutrient contents. Similarly, comparable energy contents across apps were also observed in a study where nutrient contents from the barcode scanning of 100 food products in apps were compared with product labels [31]. Likewise, Griffiths et al. compared the results of five commercial apps with thirty 24 h dietary recalls collected using the Nutrition Data System for Research (NDSR), and found a better validity of energy estimation than nutrients [33]. The mean difference of 22 kcal in energy across all apps and recipes in this study was similar with the 30 kcal mean energy difference of 23 apps compared with the three days’ weighed food record in the study of Chen et al. [14]. The wider range of energy difference (−167 to 262 kcal) in Chen’s study compared to the energy difference in our study (−118 to 141 kcal) is possibly due to a higher number of apps evaluated, and a larger amount of foods being entered in apps in Chen’s study. These findings indicated a relatively reliable energy estimation for both generic and branded food items in the current apps. Still, it was noteworthy that the largest difference of around 345 kcal between apps from both studies could impact the accuracy on both individual and population nutrient intake estimations. A trend of underestimation of energy and macronutrient contents in apps compared to reference in our study was consistent with the study by Griffiths et al. The reason in the study of Griffiths was because the food preparation details were captured by the reference (NDSR), but not in the apps. By contrast, in our study, the food details were equally captured by both the reference and apps, and the reporting bias by participants did not exist since the foods were being entered by researchers. Hence, the main reason of underestimation is the inaccuracy of the nutrition databases within the apps. 

A proper way of calculating the nutrient contents within a recipe requires the consideration of nutrient loss during cooking. Currently, the nutrient retention for foods based on different cooking processes is not calculated automatically in any dietary assessment tools, and none of the apps had instructions on using the recipe function. Although existing recipes in food composition tables take the nutrient loss into account, none of the food composition databases cover all the variations on recipes made individually [14]. Alternatively, cooked ingredients could be chosen from the food list. However, the availability of cooked ingredients was incomplete, and this would also require participants to know the amount of the prepared ingredients (which might be smaller due to shrinkage during preparation). Hence, we entered ingredients as raw ingredients, as that is the most logical option for a user.

This is the first study to investigate the discrepancies of nutrient content between raw ingredients in different apps, compared to a more accurate estimation that takes the nutrient loss into account. Only three out of twelve apps had comprehensive micronutrient information, with both actual amounts and percentage of RDA. The large variation in micronutrient content found in this study implied the importance of choosing the right nutrient database, especially when micronutrient intake estimation is part of the study purposes. The input of raw ingredients potentially leads to overestimation of several heat-sensitive micronutrients, which was shown in the micronutrient comparison between NEVO with the reference method in this study. Moreover, the results showed that the extent of difference depends largely on the nutrient contents in the recipe. Therefore, it was suggested that retention factors are most influential when applied to recipes with high micronutrient contents (e.g., boerenkool stamppot). 

NEVO was chosen as the reference measure for nutrient estimations, which was a well-maintained food composition database that had all the data on the nutrition values that were assessed and has a standardized food-compiling procedure that follows the guidelines set by EuroFIR [36,37]. Retention factors applied in this study were the most up-to-date values from the harmonization of retention factors provided by 17 EuroFIR partners [38]. However, the results of nutrient differences may lack representativeness in this study, due to a limited recipe selection. To develop a full picture of the importance of recipe calculation, additional studies, that include more recipes and an evaluation on their contribution to population nutrient intake, will be needed. Furthermore, the evaluation was done only from a research perspective in this study, while user perspective was not analyzed for the apps. Especially factors that could affect the individual’s ability to accurately enter the recipe consumed were not examined. Further development of an app for large nutrition monitoring studies would benefit from an evaluation on app users’ perspectives. 

## 5. Conclusions 

In popular food diary apps, the quality of recipe functions is suboptimal from a research perspective. All apps follow a basic nutrition-calculating algorithm, without taking nutrient retention into consideration. This leads to inaccurate nutrient intake estimations in the case that recipes are an important source of micronutrients which are vulnerable to the effects of food processing. Moreover, across apps, there is large variability in nutrient databases. From a research perspective and out of interest regarding micronutrient intake, a balance between user-friendliness and completeness of the recipe function is important. In order to obtain more insight into the need for more complex recipe functionalities, further studies on their potential impact on the nutrient intake estimations in large nutrition-monitoring studies and users’ perspective are needed.

## Figures and Tables

**Figure 1 nutrients-11-00200-f001:**
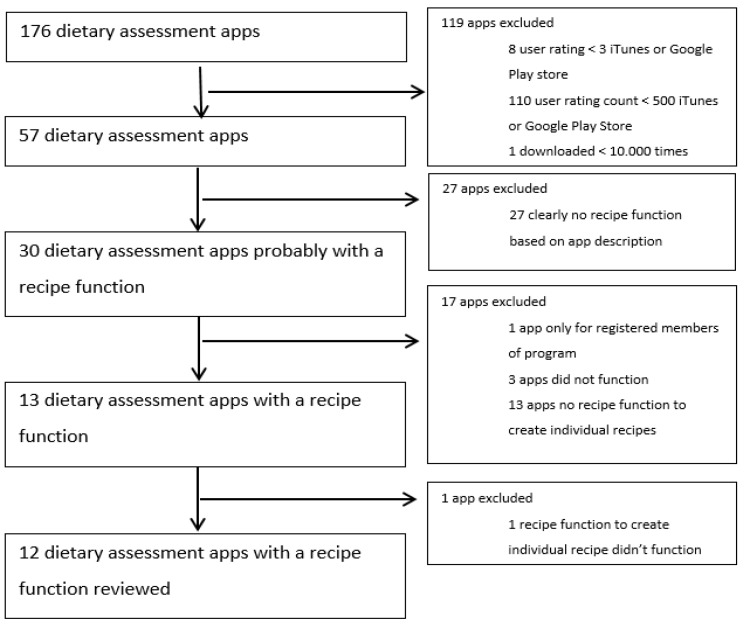
Flow diagram of selection procedure of dietary assessment apps with recipe function showing the number of apps included or excluded.

**Figure 2 nutrients-11-00200-f002:**
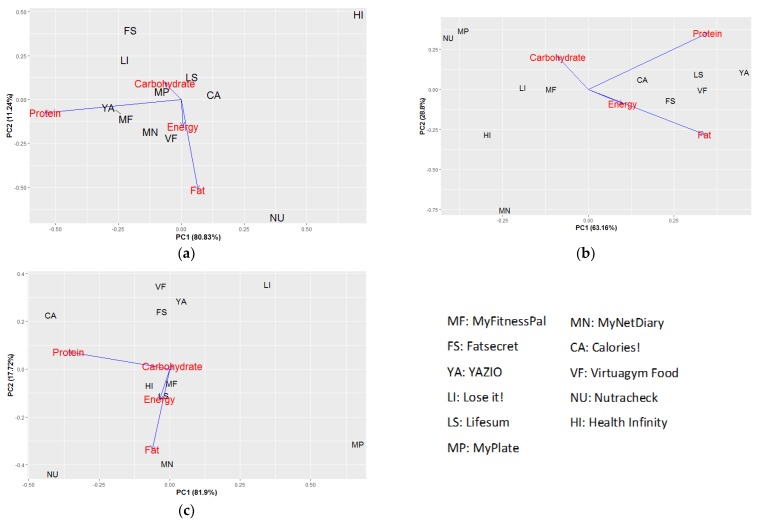
Principal Component Analysis (PCA) showing the variation strength and trend of macronutrient difference compared with the reference contents from different apps in (**a**) Boerenkool stamppot, (**b**) Pizza with salami, tomato, and mushrooms, (**c**) Hachee.

**Table 1 nutrients-11-00200-t001:** Rubric for assessment of the individual recipe function in dietary assessment apps, giving a score between 1 (low) and 5 (high) per feature.

Feature	Mark for Feature				
	1	2	3	4	5
Recipe creation options (name, photo, ingredients, servings)	The user can only create a recipe by adding ingredients and amounts	The user can create a recipe by giving it a name and adding ingredients and amounts	The user can create a recipe by giving it a name, adding ingredients and amounts, and number of servings.	The user can create a recipe by giving it a name, add ingredients and amounts, number of servings, and explanation of preparation.	The user can create a recipe by giving it a name, add ingredients and amounts, number of servings, explanation of preparation, and a photo.
Ingredients search options within a recipe	Can only search in one way	Can search in 2 or 3 ways	Can search in 4 or 5 ways	Can search in 6 or 7 ways	Can search in 8 or more ways
Reminders for frequently forgotten ingredients (e.g., olive oil, butter, salt)	App does not give reminders for frequently forgotten ingredients				App gives reminders for frequently forgotten ingredients
Preparation indication of ingredients	It is unclear whether the entered ingredients are prepared or not	The user can only select the prepared or the unprepared ingredient from the food list	The user can select both the prepared and the unprepared ingredients from the food list for some foods	The user can select both the prepared and the unprepared ingredients from the food list for all foods	The user can select an ingredient and indicate the preparation (unprepared, prepared (cooked, grilled, etc.))
Entering consumed amount at recipe level	User cannot indicate consumed amount	User can indicate the consumed amount, but the type of indication of the amount is inappropriate	User can indicate the consumed amount, and the appropriate type(s) of indication is given. However, inappropriate amounts are also given	User can indicate consumed amount and the appropriate type(s) of indication are given	User can indicate the consumed amount and the user can choose from a lot of appropriate types of indications (grams, portion in grams, portion as photo, fraction of recipe) OR can manually add amount indications
Entering prepared amount at ingredient level	User cannot indicate prepared amount	User can indicate the prepared amount, but the type of indication of the amount is limited (1 or 2 options) OR other types of indications (portion in grams, portion as photo, fraction of recipe)	User can indicate prepared amount from more than 2 options.	User can indicate prepared amount from more than 2 options. AND other types of indications (portion in grams, portion as photo, fraction of recipe)	User can indicate prepared amount from more than 2 options, and other types of indications (grams, portion in grams, portion as photo, fraction of recipe), and can manually add amount indications
Save and edit function for recipe	The user can create recipe, but cannot save it to use it later	The user can create a recipe and save it to use it later	The user can create a recipe and save it in a categorized way OR the user can create a recipe and edit it; premium only	The user can create a recipe and edit it later	The user can save the created recipe to use it later, edit it later on, and can save it in a categorized way
Energy and macronutrient information at recipe level	Energy and macronutrient content are not shown	Energy content is shown in kcal (KJ), macronutrient content is not shown	Energy content is shown in kcal (KJ), macronutrient content is shown in grams OR energy is shown in % of Reference Daily Allowance (RDA) *; premium only	Energy content is shown in kcal (KJ) and % of RDA, macronutrient content is shown in grams OR macronutrient content is shown in grams and % of RDA; premium only	Energy content is shown in kcal (KJ) and % of RDA, macronutrient content is shown in grams and % of RDA
Micronutrient information at recipe level	No micronutrient information available	Micronutrient information exists for only premium account	Information on less than 3 micronutrients	Information on 3–6 micronutrients	Information on more than 6 micronutrients

* Reference Daily Allowance (RDA): The average daily dietary intake level sufficient to meet the nutrient requirement (for the specified indicator of adequacy) of nearly all (97% to 98%) healthy individuals in a particular life stage and gender group [29].

**Table 2 nutrients-11-00200-t002:** General characteristics, such as platforms available, number of installs on Google Play Store, user rating on Google Play Store and country of twelve popular dietary assessment apps with a recipe function (*n* = 12).

	App Name (Version)	Platforms	Installs Google Play Store (Million)	Rating Google Play Store (The number of Ratings/1000)	Country
1	MyFitnessPal (18.6.0)	Android, IOS, Windows Phone	50–100	4.6 (1844)	USA
2	FatSecret (7.8.27)	Android, IOS, Windows Phone, Watch OS, Blackberry OS	10–50	4.4 (223)	Australia
3	YAZIO (4.0.1)	Android, IOS	5–10	4.6 (109)	Germany
4	Lose It! (9.4.5)	Android, IOS	5–10	4.4 (68)	USA
5	Lifesum (6.2.4)	Android, IOS, Watch OS, Android Wear	5–10	4.4 (165)	Sweden
6	MyPlate (3.2.2)	Android, IOS, Watch OS	1–5	4.6 (22)	USA
7	MyNetDiary (6.4.7)	Android, IOS, Watch OS	1–5	4.5 (26)	USA
8	Calories! (8.1.6)	Android	1–5	4.3 (10)	Germany
9	The Secret of Weight (2.4.24)	Android, IOS	1–5	4.3 (14)	France
10	Virtuagym Food (2.4.0)	Android, IOS	1–5	4.5 (28)	The Netherlands
11	Health Infinity (HI) (2.0.58)	Android	0.1–0.5	4.2 (9)	India
12	Nutracheck (5.0.12)	Android, IOS	0.1–0.5	4.3 (2)	UK

**Table 3 nutrients-11-00200-t003:** Agreed scores for the recipe function of 12 popular dietary assessment apps using the criteria list based on a 1(low)–5 (high) scale.

	App Name (Version)	MyFitnessPal (18.6.0)	FatSecret (7.8.27)	YAZIO (4.0.1)	Lose It! (9.4.5)	Lifesum (6.2.4)	MyPlate (3.2.2)	MyNetDiary (6.4.7)	Calories! (8.1.6)	The Secret of Weight (2.4.24)	Virtuagym Food (2.4.0)	Health Infinity (HI) (2.0.58)	Nutracheck (5.0.12)	Mean	SD
Criteria List															
Options (name, photo, ingredients, servings)		3	5	5	3	4	2	5	4	5	2	2	3	3.6	1.2
Options to search ingredients		2	2	3	3	3	2	3	3	2	2	2	4	2.6	0.6
Reminders for frequently forgotten ingredients (e.g., oil, spices, salt)		1	1	1	1	1	1	1	1	1	1	1	1	1.0	0.0
Entering ingredients—preparation indication		4	3	3	4	3	3	3	4	2	4	4	3	3.3	0.6
Consumed amount recipe level		4	4	4	4	4	2	4	4	4	5	1	4	3.7	1.0
Consumed amount ingredient level		3	3	3	3	3	3	3	5	2	3	2	3	3.0	0.7
Save and edit		4	5	5	4	4	4	3	4	4	4	5	5	4.3	0.6
Energy and macronutrient expression at recipe level		4	4	3	3	3	2	3	5	2	3	3	3	3.2	0.8
Micronutrient availability at recipe level		4	3	3	1	2	1	5	5	1	5	1	1	2.7	1.6
Mean		3.2	3.3	3.3	2.9	3.0	2.2	3.3	3.9	2.6	3.2	2.3	3.0	3.0	0.5
SD		1.0	1.2	1.2	1.1	0.9	0.9	1.2	1.2	1.3	1.3	1.3	1.2	0.9	-

**Table 4 nutrients-11-00200-t004:** Difference in energy (kcal) and macronutrient content (gram) estimations for one portion of each of three recipes between 12 dietary assessment apps and reference values using NEVO.

Recipes	Macronutrients	NEVO ^a^	MyFitness Pal	FatSecret	YAZIO	Lose It!	Lifesum	MyPlate	MyNet Diary	Calories!	The Secret of Weight	Virtuagym Food	Nutra Check	HI	Mean	SD
Boerenkool stamppot	Energy (kcal)	472	4	−42	10	−16	−69	−28	−53	−93	−116 *	−62	59	−44	−38	46
	Fat (g)	10.9	−0.2	−5.1 *	−0.4	−3.7 *	−1.0	−0.6	−0.2	−0.9	-	0.9	6.6 *	−2.9	−0.7	2.9
	Protein (g)	17.0	0.1	0.4	0.8	0.9	−5.2 *	−1.7	−0.2	−5.3 *	-	−1.9	−11.1 *	−17.0 *	−3.6 *	5.4
	Carbohydrate (g)	70.4	−0.1	0.3	11.8	1.2	−2.9	10.2	−15.1 *	−14.1 *	-	−11.4	−9.0	−6.1	−3.2	8.6
Pizza with salami, tomato, and mushroom	Energy (kcal)	483	−36	−5	−2	−42	−5	−35	0	−24	−7	−8	−47	−41	−21	17
	Fat (g)	25.9	−2.6	−0.3	0.3	−2.9	−0.3	−4.4 *	−0.7	−1.6	-	−0.1	−5.4 *	−2.9	−1.9	1.8
	Protein (g)	22.1	−2.3	−1.2	−0.2	−2.7 *	−0.8	−2.6 *	−5.1 *	−1.0	-	−0.9	−2.6 *	−3.8 *	−2.1	1.4
	Carbohydrate (g)	38.8	0.1	0.6	0.3	1.9	1.9	11.8	−0.8	−2.8	-	−0.4	4.2	−2.8	1.3	3.9
Hachee	Energy (kcal)	316	15	−43	−47	−119 *	7	12	75	32	58	−46	142 *	19	9	65
	Fat (g)	17.9	2.2	−4.3 *	−4.5 *	−8.8 *	2.5	1.7	8.4 *	−0.3	-	−5.1 *	10.8 *	2.4	0.4	5.6
	Protein (g)	23.3	−0.9	−0.6	−1.0	−11.2 *	−0.8	−21.3 *	−1.3	12.5 *	-	1.3	9.0 *	−0.1	−1.3	8.5
	Carbohydrate (g)	13.7	1.7	3.8	3.7	3.8	−0.9	2.3	−4.7	−4.1	-	−0.5	3.1	−1.7	0.6	3.0

^a^ Energy and macronutrient contents of one recipe portion by adding nutrient contents of raw ingredients derived from Dutch food composition database (NEVO); retention factors were all 1. * Discrepancy with reference >5% of the Dietary Reference Intakes (DRI), which is 100 kcal out of 2000 kcal for energy, 3.5 g out of 70 g for fat, 2.5 g out of 50 g for protein, and 13 g out of 260 g for carbohydrate.

**Table 5 nutrients-11-00200-t005:** Comparison of micronutrient contents between recipes added by raw ingredients from three apps with recipes added by raw ingredients from the Dutch food composition database (NEVO), with NEVO multiplied by retention factors.

					MyNetDiary			Calories!			Virtuagym		
Recipes	Micronutrients	NEVO ^a^	R ^b^	NEVO-R	App	App-NEVO	App-R	App	App-NEVO	App-R	App	App-NEVO	App-R
Boerenkool stamppot	Calcium(mg)	494	494	0	431	–63 *	–63 *	573	80 *	80 *	391	–102 *	–102 *
	Vitamin C(mg)	294	187	107 *	327	33 *	140 *	327	33 *	140 *	362	68 *	174 *
	Vitamin A(µg)	1774	1606	168 *	2557	783 *	951 *	2320	546 *	714 *	74	–1701 *	–1532 *
	Vitamin B1(mg)	0.66	0.60	0.06 *	0.32	–0.34 *	–0.28 *	0.57	–0.09 *	–0.03	0.49	–0.17 *	–0.11 *
	Vitamin B2(mg)	0.43	0.41	0.02	0.56	0.13 *	0.15 *	0.79	0.36 *	0.38 *	0.48	0.05	0.07
	Vitamin B6(mg)	1.49	1.34	0.15 *	1.38	–0.11 *	0.04	1.70	0.21 *	0.36 *	1.30	–0.19 *	–0.04
	Vitamin B12(µg)	0.11	0.11	0.00	0.43	0.32 *	0.32 *	-	-	-	0.19	0.08	0.08
	Folate(µg)	198	142	56 *	407	208 *	265 *	94	–104 *	–48 *	-	-	-
Pizza	Calcium(mg)	339	339	0	293	–46	–46	290	–48	–48	293	–46	–46
	Vitamin C(mg)	6	5	1	5	–1	3	8	2	3	5	–1	0
	Vitamin A(µg)	188	183	5	205	17	22	204	17	22	97	–91 *	–86 *
	Vitamin B1(mg)	0.21	0.18	0.03	0.75	0.54 *	0.57 *	0.29	0.08 *	0.11 *	0.77	0.56 *	0.59 *
	Vitamin B2(mg)	0.31	0.30	0.01	0.62	0.31 *	0.32 *	0.38	0.07	0.08 *	0.62	0.31 *	0.32 *
	Vitamin B6(mg)	0.26	0.24	0.02	0.27	0.01	0.03	0.31	0.05	0.07	0.26	0.00	0.02
	Vitamin B12(µg)	1.10	1.01	0.09	1.00	–0.10	–0.01	-	-	-	1.02	–0.08	0.01
	Folate(µg)	92	67	24 *	129	38 *	62 *	45	–47 *	–23 *	77	–15	10
Hachee	Calcium(mg)	51	51	0	66	15	15	48	–3	–3	67	16	16
	Vitamin C(mg)	6	5	1	8	2	3	7	1	3	9	3	4
	Vitamin A(µg)	136	129	7	108	–28	–21	123	–13	–6	94	–42 *	–34
	Vitamin B1(mg)	0.10	0.06	0.04	0.19	0.09 *	0.13 *	0.16	0.06 *	0.10 *	0.19	0.09 *	0.13 *
	Vitamin B2(mg)	0.19	0.19	0.00	0.21	0.02	0.02	0.24	0.05	0.05	0.25	0.06	0.06
	Vitamin B6(mg)	0.39	0.20	0.19 *	0.73	0.34 *	0.53 *	0.34	–0.05	0.14 *	0.73	0.34 *	0.53 *
	Vitamin B12(µg)	2.95	1.46	1.49 *	2.70	–0.25 *	1.24 *	-	-	-	2.69	–0.26 *	1.23 *
	Folate(µg)	28	15	13	57	29 *	42 *	29	1	14	13	–15	–2
The number of differences >5% DRI				8		15	14		10	11		12	10
The number of positive differences				8		11	12		7	9		5	7

^a^ Micronutrient contents of one recipe portion by adding nutrient contents of raw ingredients derived from Dutch food composition database (NEVO). ^b^ The reference measure where retention factors (RF) were multiplied by each micronutrient content derived from NEVO. * Discrepancy with reference >5% of the Dietary Reference Intake (DRI) which is 5 mg for vitamin C, 49 mg for calcium, 35 µg for vitamin A, 0.06 mg for vitamin B1, 0.08 mg for vitamin B2, 0.08 mg for vitamin B6, 0.20 µg for vitamin B12, and 17 µg for folate.

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
