# Peer review of "Evaluation of the Recipe Function in Popular Dietary Smartphone Applications, with Emphasize on Features Relevant for Nutrition Assessment in Large-Scale Studies"

_nutrients, 2019, doi:10.3390/nu11010200_

Reviewer 1 Report

The manuscript by Zhang and colleagues describes the recipe functionality of popular available food diary apps which is intended to make an ingredients inventory of different foods and dishes, and uses a research perspective to assess their accuracy and usefulness especially with regard to nutrient calculation.

In my opinion, the topic is of interest, since despite the progress in Information and Communication Technology (ICT) and the use of mobile applications (apps) on smartphones for a dietary record not much data exists concerning the accuracy of features of available recipe functions and in consequence their suitability for dietary assessment and/or for large nutrition monitoring studies.

However, after careful analysis of the manuscript, I have a few specific comments:

In the introduction, I propose to include the following reference: Food and Agriculture Organization. Dietary Assessment: A resource guide to method selection and application in low resource settings. Rome. Italy. Available online: http://www.fao.org/3/i9940en/I9940EN.

Table 1 and 2 - lack of consistency of assessment criteria. Please consider applying the same list of criteria in Table 1 as in Table 2. Is the title of table 1: Rubric for assessment ... intended? In addition, table 1 needs improvement (part 2, page 6).

In Table 5, and in description of the results I suggest to reorder micronutrients - calcium before vitamin C.

Improvements also require supplementary tables

Author Response

Point 1: In the introduction, I propose to include the following reference: Food and Agriculture Organization. Dietary Assessment: A resource guide to method selection and application in low resource settings. Rome. Italy. Available online: http://www.fao.org/3/i9940en/I9940EN.

Response 1: We thank the reviewer for suggesting this useful reference, we added this reference in line 56. 

Point 2: Table 1 and 2 - lack of consistency of assessment criteria. Please consider applying the same list of criteria in Table 1 as in Table 2. Is the title of table 1: Rubric for assessment ... intended? In addition, table 1 needs improvement (part 2, page 6).

Response 2: We apologize for the formatting error in Table 1. We have fixed the error in Table 1 in line 148. The criteria list is now complete and consistent with Table 3. The title of Table 1: Rubric for assessment....is intended, it describes the criteria list correctly. 

Point 3: In Table 5, and in description of the results I suggest to reorder micronutrients - calcium before vitamin C. Improvements also require supplementary tables

Response 3: We reordered calcium and vitamin C in both Table 5 and supplementary tables. 

Reviewer 2 Report

Your research provides a useful contribution to the literature of nutrition apps and their application.  Well done for including a systematic method to consider nutrient losses associated with cooking in your analysis.  I had several comments, many of which are typographical - the full paper would benefit from another close proofread by someone whose first language is English before resubmission.

There are some English language issues that should be addressed: eg. line 47 - increased extremely.  What does this mean?; line 57, feature should be features; line 204 micronutrient should be in plural.

Line 78 - insert "the identification of" after resulted in

Dietitians should be spelled with a t not a c, consistent with BDA, AND and DAA spelling.

Table 1 - alignment over 2 pages is distorted.

Line 156 - After manually checked should read "after manually checking for..."

Text lines 212-213 should be moved to methods and reframed as how macronutrients were managed in the analysis.

Check spelling of principal throughout.  I suggest that principal rather than principle should be used eg lines 225, 239, 144 etc.

Line 265 "did not change much" is not scientific - please rephrase.

Table 5 please add units to Line 2 of the tables.

Discussion - 2nd sentence is incomplete

line 335 users should not use an apostrophe.

line 348 - what is "a good consistency"? Please rephrase.

Some typographical errors in the reference list (particularly capitalisation).

Author Response

Point 1: There are some English language issues that should be addressed: eg. line 47 - increased extremely.  What does this mean?; line 57, feature should be features; line 204 micronutrient should be in plural.

Response 1: We thank the reviewer for the suggestions. We changed line 47 into the following :"In the last decade, an increase in the number of smartphone users has led to a proliferation of mobile applications (apps) ". We also changed feature to features in line 57, micronutrient to micronutrients in line 204.

Point 2: Line 78 - insert "the identification of" after resulted in...Dietitians should be spelled with a t not a c, consistent with BDA, AND and DAA spelling.

Response 2: We inserted "the identification of" in line 77 and changed “dieticians” to “dietitians” in line 95.

Point 3: Table 1 - alignment over 2 pages is distorted.

Response 3: Table 1 has been corrected. 

Point 4: Line 156 - After manually checked should read "after manually checking for..."

Response 4: "manually checked" has been changed to"after manually checking for" in line 156. 

Point 5: Text lines 212-213 should be moved to methods and reframed as how macronutrients were managed in the analysis.

Response 5: Line 212 has been changed to :"Macronutrient contents for both recipes and ingredients were not available in The Secret of Weight." In methods, the sentence in line 115 describes how macronutrients were managed :"For those apps that their nutrient contents were not shown at the recipe level, values from ingredients of a recipe were added up by researchers."

Point 6: Check spelling of principal throughout.  I suggest that principal rather than principle should be used eg lines 225, 239, 144 etc.

Response 6: We have changed "principle" to "principal" throughout the manuscript, in lines 141, 143, 225 and 239. 

Point 7: Line 265 "did not change much" is not scientific - please rephrase.

Response 7: We have changed line 264 into:"The proportions of relevant differences found after comparing the apps to NEVO with or without applying retention factors were rather similar (49% vs 51%)," 

Point 8: Table 5 please add units to Line 2 of the tables.

Response 8: We were not sure if we understand this point. There were units next to each nutrient in the table. Are those the units you are referring to? 

Point 9: Discussion - 2nd sentence is incomplete

Response 9: We combined the first and second sentence into:"The current study evaluated the recipe function that was available in only one fifth of the popular available food diary apps."

Point 10: line 335 users should not use an apostrophe

Response 10: We changed "user's" into "users" in line 335. 

Point 11: line 348 - what is "a good consistency"? Please rephrase.

Response 11: We rephrased the sentence in line 348 into :"Similarly, comparable energy contents across apps were also observed in a study where nutrient contents from barcode scanning of 100 food products in apps were compared with product labels."

Point 12: Some typographical errors in the reference list (particularly capitalisation).

Response 12: We checked the reference and adjusted capitalisation for journal titles.